# Melatonin Potentiates the Therapeutic Effects of Metformin in Women with Metabolic Syndrome

**Sattar J. Abood [1], Waleed K. Abdulsahib [1,*], Saad A. Hussain [2] and Sajida H. Ismail [3]**

1   Department of Pharmacology and Toxicology, College of Pharmacy, Al Farahidi University, Baghdad 10070, Iraq; lokmas14@gmail.com
2   Faculty of Pharmacy, Al-Rafidain University College, Baghdad 10064, Iraq; saad_alzaidi@yahoo.com
3   College of Pharmacy, University of Baghdad, Baghdad 10047, Iraq; sajidahusain63@yahoo.com
*   Correspondence: waleed.abdelsahib@alfarahidiuc.edu.iq

**Abstract: Objective:** This study evaluated the effect of melatonin on the response of patients suffering from metabolic syndrome (MEBS) treated with metformin. **Design:** This study used two-armed groups in a double-blind, randomized controlled clinical trial. **Materials and Methods:** A randomized double-blind placebo-controlled study was carried out on female patients diagnosed as having MEBS, according to the International Diabetes Federation (IDF) diagnosing criteria of MEBS (2005), from the outpatient clinic in Al-Zahraa Teaching Hospital/Kut, Iraq. They were diagnosed utilizing laboratory and clinical investigations, then randomized into two groups. The first group (group A) was treated with metformin (500 mg) twice daily, in addition to a placebo formula once daily at bedtime for three months. The second group (group B) was treated with metformin (500 mg) twice daily after meals, in addition to melatonin (10 mg) once daily at bedtime for three months. **Results:** The treatment of patients with MEBS using metformin–melatonin showed an improvement in most MEBS components such as fasting serum glucose (FSG), lipid profile, and body mass index (BMI), in addition to a reduction in insulin resistance and hyperinsulinemia. Simultaneously, there were increments in serum uric acid (UA), leptin, prolactin (PRL), and estradiol levels, while serum progesterone level decreased. Furthermore, patients treated with metformin–placebo showed less improvement in the studied parameters compared to that produced due to the inclusion of melatonin in the treatment protocol. **Conclusion:** Melatonin improves the effect of metformin on several components of MEBS such as FSG, lipid profile, and BMI, in addition to insulin resistance and hyperinsulinemia, compared to metformin alone.

**Keywords:** melatonin; metformin; metabolic syndrome

## 1. Introduction

Metabolic syndrome (MEBS) refers to the clustering of several cardiometabolic risk factors, including abdominal obesity, hyperglycemia, dyslipidemia, and elevated blood pressure (BP), which are linked to insulin resistance [1]. MEBS is highly prevalent and is a risk factor for cardiovascular disease (CVD), chronic kidney disease (CKD), and type II diabetes [2]. This constellation of risk factors increases the risk of diabetes by 5–9-fold and cardiovascular mortality by 2–3-fold [3]. Insulin resistance is the essential cause of MEBS [4], and it usually presents with MEBS, while it is strongly associated with other metabolic risk factors and correlates univariately with CVD risk [5]. There is a significant relationship between melatonin and insulin levels in patients with MEBS, as well as between the melatonin-insulin ratio and the lipid profile, as recently documented. Consequently, pineal disorders may be involved in prediabetes and MEBS pathogenesis, or, at a minimum, they may impact its "phenotype expression" and the intensity of its complications [6]. Metformin is primarily helpful in

obese persons with type II diabetes, a condition generally marked by insulin resistance [7]. It is also increasingly being used to treat other conditions associated with insulin resistance, especially polycystic ovary syndrome (PCOS) [8]. Melatonin (5-methoxy-*N*-acetyltryptamine) is a ubiquitous molecule which is widely distributed. It was recognized across all main classes of organism, comprising bacteria, single-cell eukaryotes, and algae, as well as in various plant parts (like seeds, flowers, stems, and roots), and vertebrate species. In vertebrates, melatonin is mainly produced and discharged by the pineal gland [9].

Some of melatonin's effects are receptor-mediated, while others are receptor-independent. Several major actions of melatonin are mediated through the activation of membrane receptors MT1 and MT2, which belong to the super family of G-protein-coupled receptors containing the typical seven transmembrane domains [9]. In mammals, melatonin has a crucial role in body mass regulation and energy expenditure [10].

Intracellular cross-talk between melatonin and insulin signaling may have a role in the intracellular mechanism controlling body weight, feeding behavior, and blood glucose circadian rhythms [10]. Insulin receptor substrates display distinct patterns of expression in the hypothalamus, pituitary gland, and olfactory bulb. Insulin's function in the hypothalamus is to inhibit feeding behavior. Melatonin encourages the activation of the insulin receptor $\beta$-subunit tyrosine kinase and fast MT1/MT2 membrane receptor-dependent tyrosine phosphorylation [11].

In addition, melatonin induces insulin receptor substrate 1 (IRS-1)/phosphatidylinositol (PI) (3) kinase and IRS-1/Src homology region 2 domain-containing phosphatase-2 (SHP-2) associations, as well as downstream protein kinase B (Akt) serine phosphorylation and P42 mitogen-activated protein kinase (MAPK) phosphorylation [12]. We suggest that IRS-1 phosphorylation may play a role as a converging target in insulin- and melatonin-stimulated signaling pathways, as well as a probable link for both hormones in the control of body weight and carbohydrate metabolism. The implication of an intracellular interaction between melatonin and insulin signaling systems is interesting. Melatonin is involved in the neuronal mechanism regulating circadian rhythms related to glucose blood level through its effects on MT1 and MT2 receptors [13].

## 2. Materials and Methods

A randomized double-blind placebo-controlled study was enrolled on 45 female patients diagnosed as having MEBS, according to the International Diabetes Federation (IDF) diagnosing criteria of MEBS (2005) [14], from the outpatient clinic in Al-Zahraa General Hospital/Kut between May 2019 and October 2019. They were diagnosed utilizing laboratory and clinical investigations. No subjects had a history of the following conditions: neurologic or psychiatric illness; allergy to melatonin or metformin; gastrointestinal motility disorder; renal or hepatic insufficiency. Patents were randomly assigned to two groups. Group A (age range: 36–66 (48.07 ± 7.43) years) was treated with metformin (500 mg twice daily), in addition to a placebo formula containing lactose in capsule dosage form for three months. Group B (age range: 27–55 (45.80 ± 6.53) years) was treated with metformin (500 mg twice daily) after meals, in addition to melatonin (10 mg in a capsule dosage form) once daily at bedtime for three months. Neither healthcare providers (physicians and nurses) nor participants were aware of the time of received medications because a third party blinded the capsule containing unlabeled medication (identified using coded numbers); thus, the medications were undisclosed to both healthcare providers and participants (patients). During the study, the patients did not modify their daily caloric intake or their physical exercise pattern. During the period of treatment, fasting serum glucose (FSG), UA, and lipid profile were evaluated before starting treatment, as well as every 30 days throughout the study. Additionally, waist circumference, body mass index (BMI), fasting serum insulin, insulin resistance index (homeostatic model assessment (HOMA) index), leptin, prolactin (PRL), estradiol, and progesterone levels were evaluated before starting treatment, as well as after 90 days of treatment.

*2.1. Statistical Analysis*

All data were statistically analyzed using paired Student's *t*-test and analysis of variance (ANOVA) to test the significant differences regarding baseline. Unpaired *t*-test was used to compare the difference between different groups (EXCEL, MICROSOFT OFFICE XP). *p* values equal or less than 0.05 was considered significantly different.

*2.2. Ethical Approval*

Approval of study procedures was obtained from Ethical Committee of the Pharmacy College, Al Farahidi University under the number of 1086 in 4/3/2019.The consent was obtained from all patients after explaining the risk and benefits of the study in their own language.

### 3. Results

Forty-five patients were randomly assigned to A or B treatment groups. Five patients (*n* = 2 from group A; *n* = 3 from group B) were withdrawn due to a history of renal disease (*n* = 3) and metformin hypersensitivity (*n* = 2). The study was performed on 40 patients. Four of these in group A declined to participate within the first 10 days. One patient in group A became pregnant, so, was excluded. Therefore, 35 patients participated in the study: 15 patients in group A were treated with metformin (500 mg twice daily) in addition to the placebo formula at bet time, and 20 patients in group B were treated with metformin (500 mg twice daily) in addition to melatonin (10 mg once daily) at bedtime. In both groups, treatment was continued for three months (Figure 1). All laboratory tests were carried out in the central laboratory of Al-Zahraa Teaching Hospital.

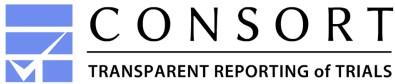

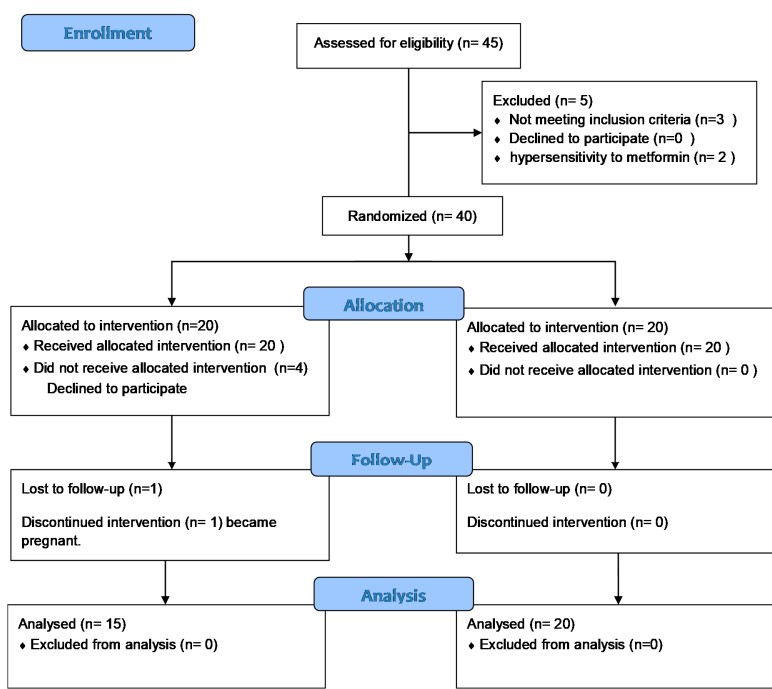

**Figure 1.** CONSORT 2010 flow diagram.

### 3.1. Effect of Treatment with Metformin-Melatonin Versus Metformin-Placebo on Fasting Serum Glucose Level in MEBS Patients

The data in Table 1 show that the treatment of women with MEBS for 90 days with metformin–melatonin resulted in significant reductions in FSG levels of 14.37% ($p = 0.0018$), 28.45% ($p = 0.0014$) and 34.63% ($p = 0.001$) after 30, 60 and 90 days, respectively, compared to pretreatment levels. Treatment with metformin–placebo also showed significant reductions ($p < 0.05$) in FBS levels of 23.68% ($p = 0.0015$), 26.82% ($p = 0.0014$) and 27.83% ($p = 0.0013$) after 30, 60, and 90 days, respectively, compared to the pretreatment levels.

**Table 1.** Effect of treatment with metformin–melatonin versus metformin–placebo on fasting serum glucose level in metabolic syndrome (MEBS) patients.

| Parameter | Duration of Treatment (Days) | Metformin (500 mg BID) + Placebo ($n = 15$) | Metformin (500 mg BID) + Melatonin (10 mg at Bedtime) ($n = 20$) |
|---|---|---|---|
| Fasting serum glucose (mmol/L) | Baseline | 7.94 ± 0.92 a | 6.96 ± 0.58 a |
| | After 30 days | 6.06 ± 0.7 * | 5.96 ± 0.49 * |
| | After 60 days | 5.81 ± 0.68 * | 4.98 ± 0.41 * |
| | After 90 days | 5.73 ± 0.66 * a | 4.55 ± 0.32 * b |

Data were presented as mean ± SEM. $n$ = number of patients. * $p < 0.05$ with respect to baseline value. Nonidentical superscripts (a, b) among different groups and within the same duration of treatment considered significantly different ($p < 0.05$). BID: twice a day.

When the effect of the two approaches of treatment on the FBS levels was compared after 90 days, the combination metformin-melatonin produces a significant ($p = 0.0012$) reduction in FSG levels when compared to the metformin-placebo group, as shown in Figure 2.

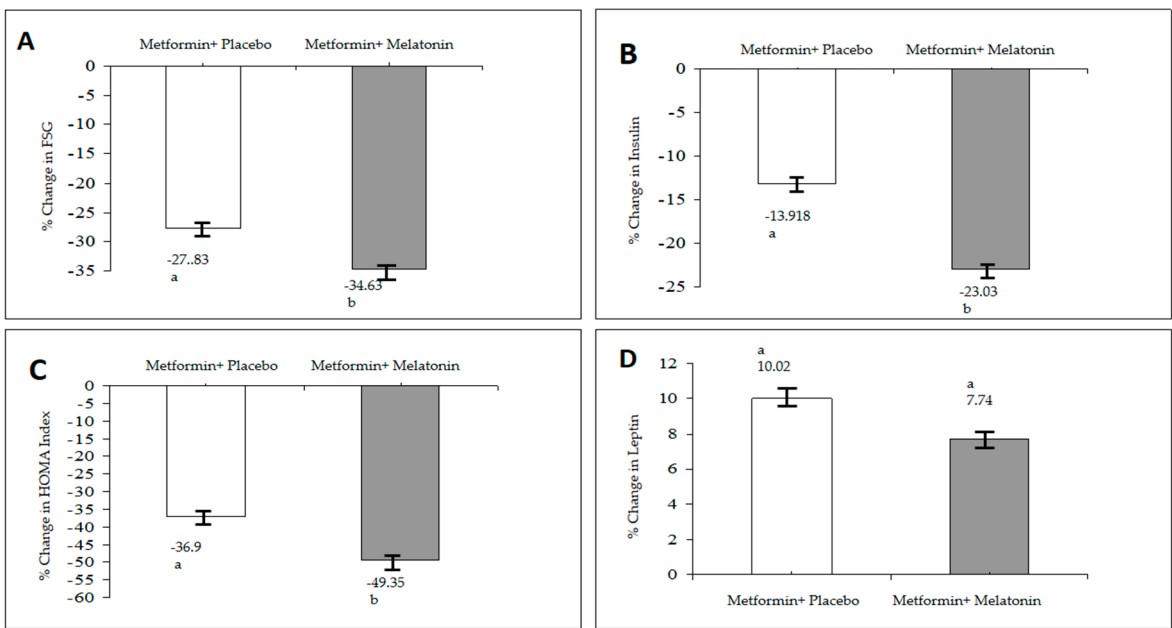

**Figure 2.** Comparison between the effects of metformin and placebo with metformin–melatonin on FSG (**A**), insulin (**B**), insulin resistance index (**C**) and on serum leptin (**D**) level in MEBS patients after 90 days of treatment; (a, b) significantly different ($p < 0.05$), (a, a) nonsignificantly different ($p > 0.05$). FSG: fasting serum glucose.

### 3.2. Effect of Treatment with Metformin-Melatonin Versus Metformin-Placebo on the Insulin Level and Insulin Resistance Based on the HOMA Index in MEBS Patients

Table 2 shows that the metformin-melatonin reduces (23%) the serum insulin level significantly ($p = 0.00173$) after 90 days in comparison with pretreatment level; this effect is also found to be significantly ($p < 0.05$) higher than that produced in patients treated with metformin-placebo (13.18%) after the same period (Figure 2).

**Table 2.** Effect of treatment with metformin–melatonin versus metformin–placebo on the serum insulin level and the insulin resistance based on the homeostatic model assessment (HOMA) index in MEBS patients.

| Parameter | Duration of Treatment (Days) | Metformin (500 mg BID) + Placebo (*n* = 13) | Metformin(500 mg BID) + Melatonin (10 mg at Bedtime) (*n* = 19) |
|---|---|---|---|
| Insulin (pmol/L) | Baseline | 121.65± 5.04 a | 120.9 ± 3.37 a |
| | After 90 days | 105.62 ± 4.18 * a | 93.09 ± 2.59 * b |
| Insulin resistance (HOMA index) | Baseline | 6.45 ± 1.15 a | 5.37 ± 0.47 a |
| | After 90 days | 4.07 ± 0.71 * a | 2.72 ± 0.22 * b |

Data were presented as mean ± SEM. *n* = number of patients. * $p < 0.05$ with respect to baseline. Nonidentical superscripts (a, b) among different group within the same duration of treatment considered significantly different ($p < 0.05$).

Table 2 also shows that the metformin–melatonin formula significantly ($p = 0.00113$) reduced the insulin resistance (a 49.41% reduction) according to the HOMA index with respect to pretreatment values, whereas the metformin–placebo reduce the insulin resistance (a 36.88% reduction) after the same period. Analysis of data after the 90th day revealed a significant difference ($p < 0.03$) in the insulin resistance between the two modes of treatment (see Figure 2).

### 3.3. Effect of Treatment with Metformin-Melatonin Versus Metformin-Placebo on the Serum Leptin Level in MEBS Patients

Table 3 shows a slight but insignificant elevation (10.02%) ($p = 0.067$) in the serum leptin level after 90 days for the group treated with a combination of metformin-placebo with respect to pretreatment level. In contrast, patients treated with a combination of metformin-melatonin revealed slightly lower and insignificant (7.74%) ($p = 0.071$) elevation after the same period of treatment. However, the difference in serum leptin levels between the two treated groups was found to be insignificant ($p = 0.09$) as shown in Figure 2.

**Table 3.** Effect of treatment with metformin–melatonin versus metformin-placebo on the serum leptin level in MEBS patients.

| Parameter | Duration of Treatment (days) | Metformin (500 mg BID) + Placebo (*n* = 13) | Metformin(500 mg BID) + Melatonin (10 mg at Bedtime) (*n* = 19) |
|---|---|---|---|
| Leptin (nmol/L) | Baseline | 5.19 ± 0.69 a | 5.30 ± 0.83 a |
| | After 90 days | 5.71 ± 1.15 a | 5.71 ± 0.84 a |

Data were presented as mean ± SEM. *n* = number of patients. (a) non-significantly different ($p > 0.05$) among different group within the same duration of treatment considered significantly different.

### 3.4. Effect of Treatment with Metformin-Melatonin Versus Metformin-Placebo on the Lipid Profile in MEBS Patients

A. Effect on serum total cholesterol (TC):

Table 4 shows a significant time-dependent reduction in serum TC levels of 14.75% ($p = 0.004$); 23.5% ($p = 0.003$); and 32.5% ($p = 0.0015$) after 30, 60, 90 days, respectively in patients treated with the metformin-melatonin combination. A similar effect, though to a lesser degree, was achieved

with the treatment by the metformin-placebo combination after 30, 60 and 90 days with values of 18% ($p = 0.02$), 21.24% ($p = 0.012$), and 26.15% ($p = 0.01$), respectively. A comparison of the percent changes in the serum TC levels at day 90 revealed that the use of metformin–melatonin was better than metformin-placebo in decreasing serum TC level ($p = 0.03$), as shown in Figure 3.

**Table 4.** Effect of treatment with metformin-melatonin versus metformin-placebo on the lipid profile in MEBS patients.

| Parameter | Duration of Treatment (Days) | Metformin (500 mg BID) + Placebo ($n = 15$) | Metformin (500 mg) + Melatonin (10 mg at Bedtime) ($n = 20$) |
|---|---|---|---|
| Serum total cholesterol (mmol/L) | Baseline | 6.31 ± 0.32 a | 6.17 ± 0.25 a |
| | After 30 days | 5.17 ± 0.26 * | 5.26 ± 0.21* |
| | After 60 days | 4.97 ± 0.25 * | 4.72 ± 0.19 * |
| | After 90 days | 4.66 ± 0.24 * a | 4.18 ± 0.17 * b |
| Serum triglyceride (mmol/L) | Baseline | 2.07 ± 0.19 a | 2.45 ± 0.21 a |
| | After 30 days | 1.67 ± 0.15 * | 1.99 ± 0.18 * |
| | After 60 days | 1.58 ± 0.14 * | 1.81 ± 0.16 * |
| | After 90 days | 1.50 ± 0.13 * a | 1.58 ± 0.14 * b |
| Serum high density lipoprotein (mmol/L) | Baseline | 0.89 ± 0.07 a | 1.08 ± 0.08 a |
| | After 30 days | 1.06 ± 0.08 * | 1.38 ± 0.10 * |
| | After 60 days | 1.28 ± 0.10 * | 1.45 ± 0.11 * |
| | After 90 days | 1.38 ± 0.11 * a | 1.72 ± 0.13 * b |
| Serum low density lipoprotein (mmol/L) | Baseline | 4.48 ± 0.32 a | 3.98 ± 0.27 a |
| | After 30 days | 3.35 ± 0.27 | 2.97 ± 0.23 * |
| | After 60 days | 2.96 ± 0.27 * | 2.44 ± 0.21 * |
| | After 90 days | 2.60 ± 0.26 * a | 1.74 ± 0.19 * b |

Data were presented as mean ± SEM. $n$ = number of patients. * $p < 0.05$ with respect to baseline value. Nonidentical superscripts (a, b) among different groups and within the same duration of treatment considered significantly different ($p < 0.05$).

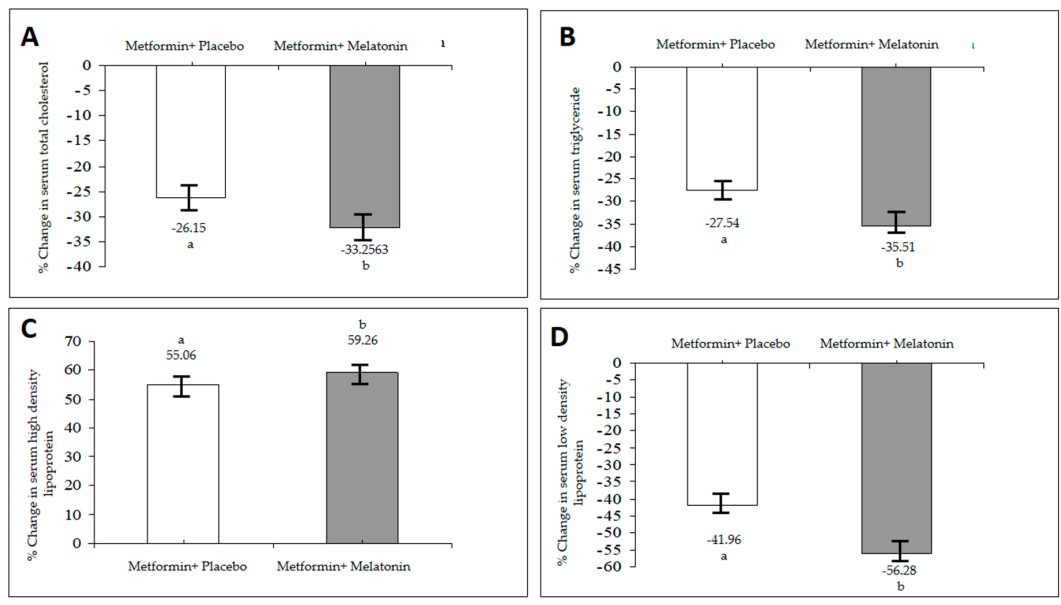

**Figure 3.** Comparison between the effects of metformin–placebo with metformin–melatonin on serum total cholesterol (TC) (**A**), triglycerides (TG) (**B**), HDL (**C**) and LDL (**D**) level in MEBS patients after 90 days of treatment, (a, b) significantly different ($p < 0.05$).

B. Effect on serum triglycerides (TG):

The data in Table 4 also show that the treatment of MEBS patients for 90 days with metformin–melatonin resulted in a significant reduction in serum TG levels: 18.78% ($p = 0.037$); 26.12% ($p = 0.0267$); 35.51% ($p = 0.015$) after 30, 60, and 90 days, respectively, regarding pretreatment levels. Treatment with metformin–placebo produced a significant reduction in serum TG levels (although to a lower degree): 19.32% ($p = 0.047$), 23.67% ($p = 0.037$) and 27.54% ($p = 0.03$) after 30, 60 and 90 days of treatment, respectively. However, there is significant difference ($p < 0.05$) in serum TG levels when comparing metformin-melatonin group with metformin- placebo group after 90 days (Figure 3).

C. Effect on serum high-density lipoprotein (HDL) cholesterol:

Treatment with metformin–melatonin resulted in a significant elevation in HDL levels: 27.28% ($p = 0.019$); 34.26% ($p = 0.011$); 59.26% ($p = 0.013$) after 30, 60 and 90 days, respectively, as shown in Table 4. On the other hand, in the metformin-melatonin group, the HDL reduction was 43.82% ($p = 0.05$) and 55.06% ($p = 0.04$) after 60 and 90 days, respectively. Analysis of data after the 90th day revealed a significant difference ($p < 0.05$) in the serum level of HDL in the two modes of treatment (see Figure 3).

D. Effect on serum low- density lipoprotein (LDL) cholesterol:

The data in Table 4 show that treatment with metformin–melatonin resulted in a significant reduction in serum LDL levels: 25.38% ($p = 0.039$); 38.69% ($p = 0.031$); and 56.28% ($p = 0.03$) after 30, 60 and 90 days, respectively, regarding pretreatment levels. In contrast, treatment with metformin–placebo showed a significant reduction regarding pretreatment levels: 33.93% ($p = 0.04$), 41.96% ($p = 0.016$) after 60 and 90 days, respectively. Analysis of data after the 90th day revealed that the combination of metformin with melatonin significantly decreased ($p < 0.05$) the serum LDL levels when compared to that produced by metformin–placebo, as shown in Figure 3.

*3.5. Effect of Treatment with Metformi-Melatonin Versus Metformin-Placebo on Serum Uric Acid Level in MEBS Patients*

As shown in Table 5, patients in the metformin-placebo group manifested a significant reduction: 8.54% ($p = 0.027$); 19.46% ($p = 0.025$); 22.35% ($p = 0.023$) in serum UA levels after 30, 60 and 90 days, respectively, while patients in the metformin–melatonin group showed a significant reduction of 18.18% ($p = 0.04$) in serum UA after 30 days, and then an increase of 10% ($p = 0.07$) in this level was observed after 90 days when compared to the pretreatment levels, as shown in Figure 4.

**Table 5.** Effect of treatment with metformin-melatonin versus metformin-placebo on serum uric acid level in MEBS patients.

| Parameter | Duration of Treatment (Days) | Metformin (500 mg BID) + Placebo (*n* = 15) | Metformin (500 mg BID) + Melatonin (10 mg at Bedtime) (*n* = 20) |
|---|---|---|---|
| Serum uric acid (μmol/L) | Baseline | 344.20 ± 29.47 a | 346.15 ± 12.28 a |
| | After 30 days | 314.82 ± 26.96 * | 283.21 ± 10.05 * |
| | After 60 days | 277.22 ± 23.74 * | 328.84 ± 11.66 * |
| | After 90 days | 267.27 ± 22.89 * a | 380.77 ± 13.51 * b |

Data were presented as mean ± SEM. *n* = number of patients. * $p < 0.05$ with respect to baseline value. Nonidentical superscripts (a, b) among different groups and within the same duration of treatment considered significantly different ($p < 0.05$).

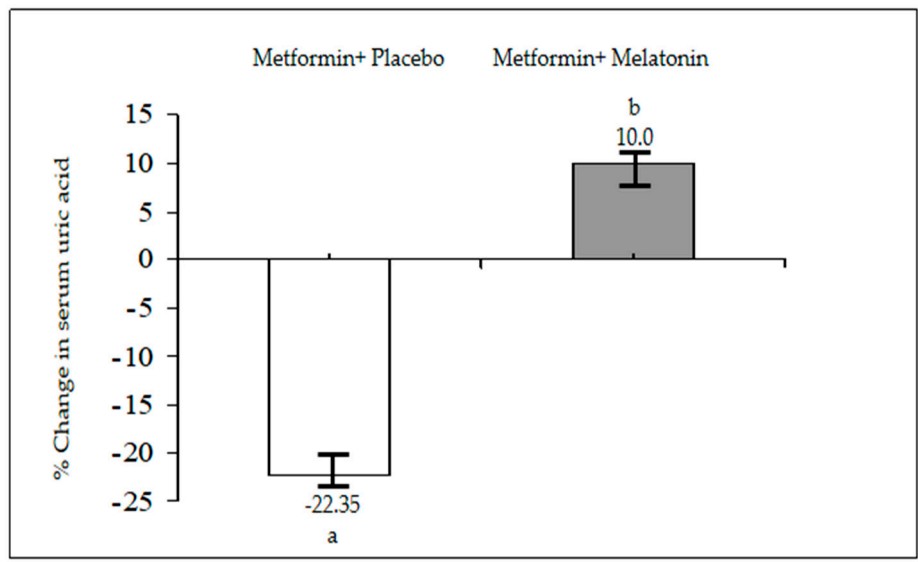

**Figure 4.** Comparison between the effects of metformin-placebo and metformin-melatonin on serum UA level in MEBS patients after 90 days of treatment; (a, b) significantly different ($p < 0.05$). UA: uric acid.

*3.6. Effect of Treatment with Metformin-Melatonin Versus Metformin-Placebo on Serum Levels of Estrogen, Progesterone, and Prolactin in MEBS Patients*

The data presented in Table 6 show that treatment of MEBS patients with the metformin-melatonin combination resulted in a significant elevation (12.5% ($p = 0.04$)) in the serum estradiol level after 90 days. However, the effect was comparable and does not significantly differ with what observed during treatment with metformin–placebo at the same period (11.12% ($p = 0.33$)), as shown in Figure 5.

**Table 6.** Effect of treatment with a combination of metformin-melatonin versus metformin-placebo on some hormones in MEBS Patients.

| Parameter | Duration of Treatment (Days) | Metformin (500 mg BID) + Placebo (*n* = 13) | Metformin (500 mg BID) + Melatonin (10 mg at Bedtime) (*n* = 19) |
|---|---|---|---|
| Estradiol (pmol/L) | Baseline | 80.60 ± 5.72 a | 88.31 ± 7.56 a |
|  | After 90 days | 89.56 ± 8.93 * a | 99.36 ± 5.42 * a |
| Progesterone (nmol/L) | Baseline | 14.42 ± 1.01 a | 15.20 ± 0.83 a |
|  | After 90 days | 13.71 ± 0.92 * a | 13.30 ± 1.11 * a |
| Prolactin (pmol/L) | Baseline | 642.21 ± 39.47 a | 617.86 ± 24.5 a |
|  | After 90 days | 393.44 ± 21.61 * a | 746.94 ± 24.50 * b |

Data were presented as mean ± SEM. *n* = number of patients. * $p < 0.05$ with respect to baseline. Nonidentical superscripts (a, b) among different group within the same duration of treatment considered significantly different ($p < 0.05$).

Concerning the effect on serum progesterone levels, Table 6 shows a viable reduction (12.5% ($p = 0.029$)) in this hormone after 90 days in the metformin-melatonin group; but this effect was nonsignificant (4.92% ($p = 0.34$)) when compared with metformin-placebo group at the same period, as shown in Figure 5.

Table 6 showed a significant elevation (20.89% ($p = 0.044$)) in the serum PRL level after 90 days in the metformin–melatonin group. On the other hand, the metformin-placebo produced a significant reduction (38.74% ($p = 0.027$)) in the serum PRL level after the same period. Upon statistical analysis, the difference was significant between the two groups ($p < 0.05$), as shown in Figure 5.

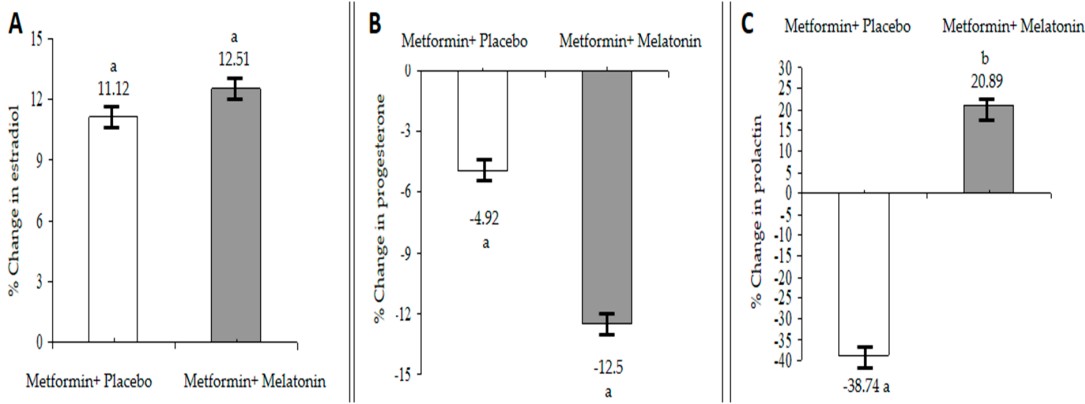

**Figure 5.** Comparison between the effects of metformin–placebo with metformin-melatonin on serum estradiol(**A**), progesterone (**B**) and prolactin (**C**) level in MEBS patients after 90 days of treatment; (a, a) non-significantly different ($p > 0.05$), (a, b) significantly different ($p < 0.05$).

*3.7. Effect of Treatment with Metformin-Melatonin Versus Metformin-Placebo on the BMI and Waist Circumference in MEBS Patients*

The data in Table 7 show that treatment with metformin–melatonin resulted in a significant reduction of 3.11% ($p = 0.0026$) in BMI after 90 days when compared to the baseline value. Meanwhile, treatment with metformin–placebo produced no significant ($p = 0.07$) changes in BMI after the same period. Waist circumference (WC) decreased significantly in both groups and there is an insignificant difference when comparing two groups after 90 days, as shown in Figure 6.

**Table 7.** Effect of treatment with metformin-melatonin versus metformin-placebo on the BMI and waist circumference (WC) in MEBS patients.

| Parameter | Duration of Treatment (days) | Metformin (500 mg BID) + Placebo (*n* = 15) | Metformin (500 mg BID) + Melatonin (10 mg at Bedtime) (*n* = 20) |
|---|---|---|---|
| BMI (kg/m$^2$) | Baseline | 41.81 ± 2.26 | 40.24 ± 1.55 |
| | After 90 days | 41.42 ± 2.22 *a | 38.99 ± 1.61 * b |
| WC (cm) | Baseline | 121.10 ± 3.66 | 115.70 ± 2.84 |
| | After 90 days | 99.67 ± 6.22 * a | 97.17 ± 4.12 * a |

Data were presented as mean ± SEM. *n* = number of patients. * $p < 0.05$ with respect to baseline. Nonidentical superscripts (a, b) among different group within the same duration of treatment considered significantly different, $p < 0.05$. BMI (body mass index), WC (waist circumference).

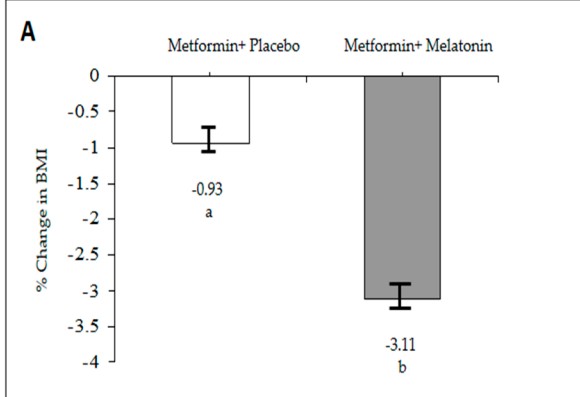
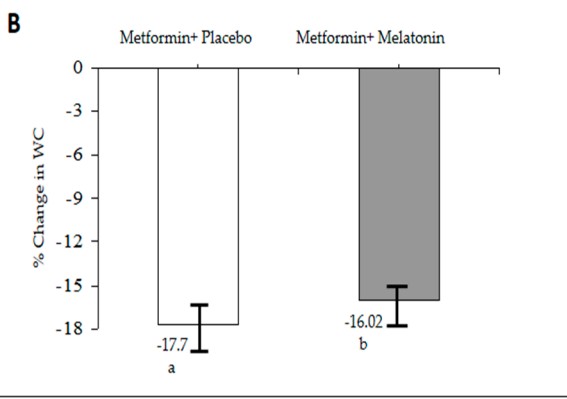

**Figure 6.** Comparison between the effects of metformin-placebo and metformin-melatonin on BMI (**A**) and WC (**B**) measurements in MEBS patients after 90 days of treatment; (a, b) significantly different ($p < 0.05$); BMI (body mass index), WC (waist circumference).

## 4. Discussion

The considerable reduction in FSG levels were obtained in the metformin-melatonin group, which may be attributed to several mechanisms proposed by several studies in this field. The carbohydrate metabolism is influenced by melatonin both in humans and rodents [15].

Recently, a study in healthy adults regarding the influences of a nocturnal lifestyle on the melatonin circadian pattern showed that the night melatonin level was diminished in the persons who preserved a nocturnal lifestyle. The fact that the nocturnal lifestyle group showed a higher glucose level than that of the diurnal lifestyle group leads to the proposal that a nocturnal lifestyle is a possible hazardous factor for diabetes [16].

It was demonstrated by in vitro studies that melatonin induces a rapid MT 1/MT 2 membrane receptor-dependent tyrosine phosphorylation as well as the activation of the insulin receptor β-subunit tyrosine kinase [17]. In addition, melatonin induces IRS-1/PI [3] kinase and IRS-1/SHP-2 associations as well as downstream Akt serine phosphorylation and extracellular signal-regulated kinase-2 (ERK-2) phosphorylation respectively in the hypothalamic suprachiasmatic nucleus of rat [17].

Melatonin and leptin play an essential function in the organization of body mass and energy balance. Both hormones display a circadian rhythm, with rising values at night [18]. The remarkable improvement in glucose tolerance after giving leptin to fat animal models, as well as the observed decrease in the level of insulin in the plasma, is due to the improvement in insulin action in vivo, stimulating glucose disposal in peripheral tissues and reducing glucose production in the liver where the main effects of insulin occur [18].

Recently, in vitro study confirmed that melatonin or insulin alone did not affect leptin expression, but together they increased it by 120%. It is proposed that melatonin acts with insulin and upregulates insulin-triggered leptin expression. Meanwhile, melatonin raises the insulin induced-insulin receptor *β*-tyrosine phosphorylation, which leads to a raised serine phosphorylation of the downstream convergent protein Akt. These effects are caused by melatonin binding to the Gi-protein-coupled MT 1 membrane receptors, although adipocytes express two types of melatonin receptors by lowering c-AMP levels [19].

In addition, an elevated glucose level in the blood causes a raise in the glycation of lipoproteins, including LDL and HDL linked to an increase in TG levels by increasing the synthesis from glucose and impaired lipid metabolism [19].

Thus, the improvement in the insulin sensitivity of melatonin may be the key factor in the resolution of the dilemma of hyperlipidemia associated with MEBS. It can be explained by the results of several studies that attempted to link these disturbances to the role of the pineal gland in regulating energy expenditure and body weight regulation. In hypercholesterolemia, the administration of melatonin reduces serum cholesterol and LDL level, and it can also enhance the catabolism of cholesterol to bile acids [20]. Furthermore melatonin can obstruct the synthesis of cholesterol and LDL accumulation [20,21]. These results recommend that melatonin has a hypocholesterolemic effect by increasing clearance mechanisms of the endogenous cholesterol [22].

Uric acid level may rise as a compensatory response to counteract the increased oxidative stress under the conditions of MEBS or atherosclerosis [23]. Although the exact mechanism is not clear, an increase in the serum UA level within a normal range has also been obtained after adjuvant treatment of uncontrolled hyperglycemia with the same dose of melatonin and zinc acetate in type I diabetic patients when compared to pretreatment values [24]. These high levels may be attributed to either increased production or a decrease in urinary elimination of UA. In addition to both melatonin metabolites and UA excreted by the kidney, there may also exist a certain type of competition between the two metabolites at the site of excretion in the renal tubule. On the other hand, an important reduction in the serum UA level appeared in the metformin–placebo group. Hyperuricemia is a marker of insulin resistance and an underlying condition of MEBS. Remarkably, the improvement of insulin resistance through diet or insulin sensitizing agents diminishes the serum UA level, which may favor the notion

that a high serum level of UA may be a part of insulin resistance syndrome [25]. Therefore, decreased levels of UA may point to the improvement in insulin resistance because of treatment with metformin.

The inverse relationship between melatonin and estrogen levels is well established [26], as it was found that exogenous melatonin decreases estradiol and increases progesterone in healthy women [27]. However, the current study showed different findings: an increase in the serum estradiol levels after 90 days of treatment with melatonin and metformin. On the other hand, the progesterone level is in agreement with the majority of studies regarding the effect of melatonin in this respect. Since the vast majority of the patients in the present study were at the pre- or post-menopausal state, more detailed hormonal follow-ups after each menstrual cycle may provide more explanations. The observed increase in the serum estradiol level may explain the unexpected role of the pineal gland in regulating estrogen produced by the ovaries. Additionally, treatment of PCOS women with metformin increases insulin sensitivity and improves the hyperinsulinemia, resulting in a decreased total, free levels of testosterone, and increased estradiol levels [28]. Therefore, our results are in agreement with these findings, and improving hyperinsulinemia may be responsible for this reported elevation in estradiol levels.

It has recently been reported that MEBS is associated with a reduced serotonergic function, since serotonin, which is a precursor of melatonin, has a role in regulating PRL release from the anterior pituitary gland [29]. Prolactin is a very versatile hormone; it shows an important role in the adjustment of carbohydrate and lipid metabolism in various species [30].

Both endogenous and exogenous melatonin enhances PRL secretion without affecting its circadian rhythm [30]. Additionally, leptin may promote PRL release in obese humans. Furthermore, central serotonergic activity, represented by PRL release, influences eating behaviors, such as thermogenesis; physical activity; circadian rhythm; pancreatic function; autonomic function [31]. Many studies supported the existence of interplay between PRL and insulin along with the influence of dopamine in regulating insulin secretion [32] as the total daily PRL release is strongly associated with the size of the visceral fat depot [33]. A decreased PRL level that accompanies the treatment with metformin may be attributed to the general decrease in BMI.

Pinealectomy, which depresses but does not eliminate circulating melatonin, has been demonstrated to increase body weight and food consumption in rats [34]. Melatonin has been found to regulate body weight because of its ability to prevent the consequences of dyslipidemia and improve glucose homeostasis [34]. Furthermore, independently of food intake and total body fat, regular melatonin administration to male rats represses body weight, abdominal adiposity, insulin and leptin release. Thus, the resulting decrease in BMI and WC in the present study can be explained according to the abovementioned effects of melatonin. Additionally, hyperinsulinemia and increased visceral fat are commonly associated with aging and a decrease in melatonin levels in both rats and humans [35], and replacement therapy with melatonin may contribute to reducing visceral obesity and decreasing BMI. Regarding the effect of metformin alone, it shows a significant decrease in both BMI and WC; this might be attributed to its effect on adipose tissues, where it reduces caloric intake [36]. However, the reduction in BMI is higher upon addition of melatonin. Treatment of obese women with PCOS with metformin resulted in body weight loss, which was further enhanced by increasing the dose that may be associated with side effects [37]. Thus, adding melatonin may produce the same or greater effect in this respect, with the possibility of reducing the dose of metformin in order to eliminate side effects.

## 5. Conclusions

Melatonin improves the effect of metformin on several components of MEBS such as FSG, lipid profile, BMI, insulin resistance and hyperinsulinemia when compared to metformin alone. Pharmacological doses of melatonin may produce an additive effect to metformin when utilized long-term in patients with MEBS, leading to possibility of reducing the dose of metformin and subsequently decreasing its side effects.

**Author Contributions:** Conceptualization, S.J.A., S.A.H. and S.H.I.; data curation, W.K.A.; manuscript preparation, S.J.A. and W.K.A.; investigation, S.A.H. and S.H.I.; methodology, S.J.A., W.K.A., S.A.H. and S.H.I.; project administration, S.A.H., S.H.I. and S.J.A.; writing—original draft, S.J.A.; writing—review and editing, W.K.A. All authors have read and agreed to the published version of the manuscript.

**Funding:** This research received no external funding.

**Acknowledgments:** Not applicable.

**Conflicts of Interest:** The authors declare that there is no conflict of interest.

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
