# Peer review of "Melatonin Potentiates the Therapeutic Effects of Metformin in Women with Metabolic Syndrome"

_scipharm, doi:10.3390/scipharm88020028_

Round 1

Reviewer 1 Report

Summary

The paper by Abbood et. al. reports randomized clinical trial findings evaluating the effect of melatonin on diabetic patients with metabolic syndromes (MEBS) and taking Metformin to manage diabetes. This is a carefully conducted clinical trial reporting several biological parameters to measure the impact of melatonin against a placebo on the MEBS. It has been known that melatonin mediated signaling has a significant effect on cellular metabolism, and the findings reported in this study supports that. While it can be challenging to achieve ideal conditions to conduct a study of this nature in order to improve high statistical power to establish the link and the effect of melatonin on MEBS, findings in this study would certainly of interest to the physicians and researchers working to improve therapeutic outcomes of diabetes; however, some points need to be addressed.

Major Points

This study reports effects of melatonin on several key parameters, however, given the size of the patients in this controlled clinical trial, it would likely require a larger sample size to make statistically significant conclusions. In the absence of such sample size, citing the actual P-value, at the least, would help understand the outcomes. While the P-value of < 0.05 indicates the significance, including the actual value, it would improve the quality of this report.

Furthermore, there are several findings, for example, the recently published 3-D crystal structures of MT1 and MT2 receptors (Stauch et al., Nature 2019, Johansson et al. Nature, 2019) discusses the structural aspect, as well as the mutations in the melatonin receptor and its impact on type-2 diabetes (Karamitri et al., Sci. Signaling, 2018). It would be vital to put those findings in the context while establishing the effects of Metformin+Melatonin in the treatment of diabetes.

Finally, discussing the cellular pathways of metformin and melatonin, along with the intersection of two, would also add value to the paper.

Minor points

It would be essential to include the error bars in the plot figures.

Author Response

Dear reviewer,

Thank you too much for your worthy notes.

with best regards,

Waleed

Reviewer 2 Report

Critical evaluation of the manuscript Satte J. Abbood et al. entitled „Melatonin potentiate the therapeutic effects of Metformin in Women with Metabolic Syndrome

 General comments:

This manuscript consists of 17 numbered pages and 38 references. The references are properly collected and arranged.

 In formal aspect it is an appropriate work, and meets the requirements determined by the journal. However, many spelling mistakes can be found in the manuscript and there is no standard format, somewhere  the line length is 1.0, somewhere it is 1.15. The authors should be improved. Stronge English revision is required.

Specific comments:

The authors made clinical trial to certify the melatonin can potentiate the metformin effect in the case if metabolic syndrome.

The definition what the authors cited from 2005 (International Diabetes Federation diagnosing criteria of MEBS) is not up-to-date, there are many new guideline about metabolic syndrome, in that case the interpretation of metabolic syndrome is debatable.

Those human clinical trial which involves 45 (after extrusion is 40) is considered as a pilot human study.

Determination of HOMA index and the leptin level are representative, but there is the lack of oral glucose tolerance test and hyperinsulin euglicemic clamp measurement. These test could be more informative if we want to certify that medicine are more effective for the treatment of insulin resistance.

Author Response

Dear reviewer, 

Thank you very much for your important notes.

Please see the attachment below.

Regards,

Waleed

Round 2

Reviewer 2 Report

I accept it .